# COMPLEMENTARY-LABEL LEARNING FOR ARBITRARY LOSSES AND MODELS

## ABSTRACT

In contrast to the standard classification paradigm where the true (or possibly noisy) class is given to each training pattern, *complementary-label learning* only uses training patterns each equipped with a complementary label. This only specifies one of the classes that the pattern does *not* belong to. The seminal paper on complementary-label learning proposed an unbiased estimator of the classification risk that can be computed only from complementarily labeled data. However, it required a restrictive condition on the loss functions, making it impossible to use popular losses such as the softmax cross-entropy loss. Recently, another formulation with the softmax cross-entropy loss was proposed with consistency guarantee. However, this formulation does not explicitly involve a risk estimator. Thus model/hyper-parameter selection is not possible by cross-validation— we may need additional ordinarily labeled data for validation purposes, which is not available in the current setup. In this paper, we give a novel general framework of complementary-label learning, and derive an unbiased risk estimator for arbitrary losses and models. We further improve the risk estimator by non-negative correction and demonstrate its superiority through experiments.

## 1 INTRODUCTION

Modern classification methods usually require massive data with high-quality labels, but preparing such datasets is unrealistic in many practical domains. To mitigate the problem, many previous works have investigated ways to learn from weak supervision: semi-supervised learning (Chapelle et al., 2006; Miyato et al., 2016; Kipf & Welling, 2017; Sakai et al., 2017; Tarvainen & Valpola, 2017; Oliver et al., 2018), learning from noisily-labeled data (Natarajan et al., 2013; Patrini et al., 2017; Ma et al., 2018), learning from positive-unlabeled data (Elkan & Noto, 2008; du Plessis et al., 2014; 2015; Kiryo et al., 2017), learning from similar-unlabeled data (Bao et al., 2018), learning from positive-confidence data (Ishida et al., 2018), and others.

In this paper, we consider learning from another type of weak but natural supervision called *complementary-label learning* (Ishida et al., 2017; Yu et al., 2018), where the label only specifies one of the classes that the pattern does *not* belong to. In contrast to the ordinary case where the true class is given to each pattern (which often needs to be chosen out of many candidate classes precisely), collecting these complementary labels is obviously much easier and less costly. A natural question is, however, is it possible to learn from such complementary labels (without any *true labels*)?

The problem has previously been tackled by Ishida et al. (2017), showing that the classification risk can be recovered only from complementarily labeled data. They also gave consistency gaurantee in theoretical analysis. However, they required strong restrictions on the loss functions, allowing only one-versus-all and pairwise comparison multi-class loss functions (Zhang, 2004) with certain non-convex binary losses. This is a severe limitation when we use deep learning since the softmax cross-entropy loss is often used to boost the classification performance.

Later, Yu et al. (2018) proposed a different formulation for complementary labels by employing the forward loss correction technique (Patrini et al., 2017) to adjust the learning objective. Their proposed risk estimator is not necessarily *unbiased* but the minimizer is theoretically guaranteed to be *consistent* with the minimizer of the risk for ordinary labels (under an implicit assumption on the model for convergence analysis).

**Table 1:** Comparison of two proposed complementary-label methods with previous works.

| Methods | loss assump. free | model assump. free | unbiased estimator | explicit risk correction |
|---|---|---|---|---|
| Ishida et al. (2017) | ✕ | ✓ | ✓ | ✕ |
| Yu et al. (2018) | ✕ | ✕ | ✕ | ✕ |
| Proposed (General formulation) | ✓ | ✓ | ✓ | ✕ |
| Proposed (Non-negative formulation) | ✓ | ✓ | ✕ | ✓ |

They also extended the problem setting to where complementary labels are chosen in an uneven (biased) way. This is a realistic problem setting because labelers are more likely to complementarily label a pattern when they feel it is not a certain class which they have more knowledge or experience about.

In this paper, we first derive an unbiased risk estimator with a general loss function, making *any* loss functions available for use: not only the softmax cross-entropy loss function but other convex/non-convex loss functions can also be applied. We also do not have implicit assumptions on the classifier, allowing both linear and non-linear models.

Yu et al. (2018) does not have an unbiased risk estimator, which means users will need clean data with true labels to calculate the error rate during the validation process. On the other hand, our proposed unbiased risk estimator can handle *complementarily* labeled validation data not only for our learning objective, but also for Yu et al. (2018). This is helpful since collecting clean data is usually much more expensive.

Finally, our proposed unbiased risk estimator has an issue that it is unbounded from below and suffers from the classification risk going to negative after learning, leading to overfitting. We further propose a non-negative correction to the original unbiased risk estimator to improve our estimator. We experimentally show that our proposed method is comparable to or better than previous methods (Ishida et al., 2017; Yu et al., 2018) in terms of classification accuracy.

## 2 REVIEW OF PREVIOUS WORKS

In this section, we explain the notations and review the formulations of learning from ordinary labels, learning from complementary labels, and learning from both ordinary and complementary labels.

**Learning from ordinary labels** Let $\mathcal{X}$ be an instance space and $\mathcal{D}$ be the joint distribution over $\mathcal{X} \times [K]$ for class label set $[K] := \{1, 2, \ldots, K\}$, with random variables $(X, Y) \sim \mathcal{D}$. The data at hand is sampled independently and identically from the joint distribution: $\{(x_i, y_i)\}_{i=1}^n \overset{\text{i.i.d.}}{\sim} \mathcal{D}$. The joint distribution $\mathcal{D}$ can be either decomposed into class-conditionals $\{P_k\}_{k=1}^K$ and base rate $\{\pi_k\}_{k=1}^K$, where $P_k := \mathbb{P}(X|Y = k)$ and $\pi_k := \mathbb{P}(Y = k)$, or the marginal $M$ and class-probability function $\boldsymbol{\eta} : \mathcal{X} \to \Delta_k$, where $M := \mathbb{P}(X)$ and $\boldsymbol{\eta}_k(x) := \mathbb{P}(Y = k|X = x)$. A loss is any $\ell : [K] \times \mathbb{R}^K \to \mathbb{R}_+$ and the decision function is any $\boldsymbol{g} : \mathcal{X} \to \mathbb{R}^K$. The risk for the decision function $\boldsymbol{g}$ with respect to loss $\ell$ and implicit distribution $\mathcal{D}$ is:

$$R(g; \ell) := \mathbb{E}_{(X,Y)\sim\mathcal{D}}[\ell(Y, \boldsymbol{g}(X))], \tag{1}$$

where $\mathbb{E}$ denotes the expectation. Two useful equivalent expressions of classification risk (1) used in later sections are

$$R(g; \ell) := \mathbb{E}_X[\boldsymbol{\eta}(x)^T \boldsymbol{\ell}(\boldsymbol{g}(X))] = \sum_{k=1}^K \pi_k \mathbb{E}_{\mathbb{P}_k}\Big[\boldsymbol{\ell}(k, \boldsymbol{g}(X))\Big], \tag{2}$$

where $\boldsymbol{\ell} := [\ell(1, \boldsymbol{g}), \ell(2, \boldsymbol{g}), \ldots, \ell(K, \boldsymbol{g})]^T$. The goal of classification is to learn the decision function $\boldsymbol{g}$ that minimizes the risk. In the usual classification case with ordinarily labeled data at hand, approximating the risk empirically is straightforward: $\widehat{R}(g; \ell) := \frac{1}{n} \sum_{i=1}^n \ell(y_i, \boldsymbol{g}(x_i))$.

**Learning from complementary labels**   Next we consider the problem of learning from complementary labels (Ishida et al., 2017). We observe patterns each equipped with a complementary label $\{(x_{i'}, \overline{y}_{i'})\}_{i'=1}^{n'}$ sampled independently and identically from a different joint distribution $\overline{\mathcal{D}} \neq \mathcal{D}$. We denote random variables as $(X, \overline{Y}) \sim \overline{\mathcal{D}}$. As before, we assume this distribution can be decomposed into either class-conditionals $\{\overline{P}_k\}_{k=1}^{K}$ and base rate $\{\overline{\pi}\}_{k=1}^{K}$, or marginal $M$ and class-probability function $\overline{\eta} : \mathcal{X} \to \Delta_K$, where $\overline{P}_k := \mathbb{P}(X|\overline{Y} = k)$, $\overline{\pi}_k := \mathbb{P}(\overline{Y} = k)$, $M := \mathbb{P}(X)$, $\overline{\eta}_k(x) := \mathbb{P}(\overline{Y} = k|X = x)$, $\overline{Y}$ is the complementary label, and $\Delta_K$ is the conditional probability simplex for $K$ classes. Without any assumptions on $\overline{\mathcal{D}}$, it is impossible to design a suitable learning procedure. The assumption for unbiased complementary learning used in Ishida et al. (2017) was

$$\overline{\eta}(x) = T\eta(x), \tag{3}$$

where $T \in \mathbb{R}^{K \times K}$ is a matrix that takes 0 on diagonals and $\frac{1}{K-1}$ on non-diagonals. Under this assumption, Ishida et al. (2017) proved that they can recover the classification risk (1) from an alternative formulation using only complementarily labeled data when he loss function satisfies certain conditions. More specifically, usable loss functions are pairwise comparison or one-versus-all multi-class loss functions (Zhang, 2004) each with binary loss function $\ell'(z) : \mathbb{R} \to \mathbb{R}_+$ that satisfies $\ell'(z) + \ell'(-z) = 1$, such as ramp loss $\ell'_R(z) = \frac{1}{2} \max\left(0, \min(2, 1 - z)\right)$ or sigmoid loss $\ell'_S(z) = \frac{1}{1+e^z}$.

Having an unbiased risk estimator is also helpful for the validation process. Since we do not have ordinary labels in our validation set in the complementary-label learning setting, we cannot follow the usual validation procedure that uses zero-one error or accuracy. If we have an unbiased estimator of the original classification risk (which can be interpreted as zero-one error), we can use the empirical risk for (cross)-validated complementary data to select the best hyper-parameter or deploy early stopping.

An extension of the above method was considered in Yu et al. (2018) by using a different assumption than the unbiased complementary learning of Ishida et al. (2017): there is some bias amongst the possible complementary labels that can be chosen, thus the non-diagonals of $T$ is not restricted to $\frac{1}{K-1}$. However, one will need to prepare a separate dataset with ordinary labels in order to estimate $T$ beforehand.

Unlike Ishida et al. (2017), Yu et al. (2018) did not directly provide a risk estimator, but they showed that the *minimizer* of their learning objective agrees with the minimizer of the original classification risk (1). Note that, in their formulation, the loss function is restricted to the softmax cross-entropy loss. Furthermore, the use of a highly non-linear model is supposed for consistency guarantee in their theoretical analysis. Since the learning objective of Yu et al. (2018) does not correspond to the classification risk, one will need clean data with true labels to calculate the error rate during the validation process. On the other hand, our proposed risk estimator can cope with *complementarily* labeled validation data not only for our own learning objective, but can be used to select hyper-parameters for others such as Yu et al. (2018).

**Learning from both ordinary and complementary labels**   In many practical situations, we may also have ordinarily labeled data in addition to complementarily labeled data. Ishida et al. (2017) touched on the idea of crowdsourcing for an application with both types of data. For example, we may choose one of the classes randomly by following the uniform distribution, with probability $\frac{1}{K-1}$ for each class, and ask crowdworkers whether a pattern belongs to the chosen class or not. Then the pattern is treated as ordinarily labeled if the answer is yes; otherwise, the pattern is regarded as complementarily labeled. If the true label was $y$ for a pattern, we can naturally assume that the crowdworker will answer yes by $\mathbb{P}(Y = y|X = x)$ and no by $1 - \mathbb{P}(Y = y|X = x)$. This way, ordinarily labeled data can be regarded as samples from $\mathcal{D}$, and complementarily labeled data from $\overline{\mathcal{D}}$, justifying the assumption of unbiased complementary learning (3). In Ishida et al. (2017), they considered a convex combination of the classification risks derived from ordinarily labeled data and complementarily labeled data: $\alpha R(g; \overline{\ell}) + (1 - \alpha)R(g; \ell)$, where $\alpha \in [0, 1]$ is a hyper-parameter that interpolates between the two risks. The combined (also unbiased) risk estimator can utilize both kinds of data in order to obtain better classifiers, which was demonstrated to perform well in experiments.

## 3 PROPOSED METHOD

As discussed in the previous section, the method by Ishida et al. (2017) works well in practice, but it has restriction on the loss functions—the popular softmax cross-entropy loss is not allowed. On the other hand, the method by Yu et al. (2018) allows us to use the softmax cross-entropy loss, but it does not directly provide an estimator of the classification risk and thus model selection is problematic in practice. We first describe our general unbiased risk formulation in Section 3.1. Then we discuss how the estimator can be further improved in Section 3.2. Third, we propose a way for our risk estimator to avoid overfitting by a *non-negative risk estimator* in Section 3.3. Finally, we show practical implementation of our risk estimator with stochastic optimization methods in Section 3.4.

### 3.1 GENERAL RISK FORMULATION

First, we describe our general unbiased risk formulation. We give the following theorem, which allows unbiased estimation of the classification risk from complementarily labeled samples:

**Theorem 1.** *For any ordinary distribution $\mathcal{D}$ and complementary distribution $\overline{\mathcal{D}}$ related by (3) with decision function $\boldsymbol{g}$, and loss $\ell$, we have*

$$R(\boldsymbol{g}; \ell) = R(\boldsymbol{g}; \overline{\ell}) \tag{4}$$

*for the complementary loss*

$$\overline{\boldsymbol{\ell}}(\boldsymbol{g}) := \Big( -(K-1)\boldsymbol{I}_K + \frac{1}{K-1}\boldsymbol{1}\boldsymbol{1}^\top \Big) \cdot \boldsymbol{\ell}(\boldsymbol{g}), \tag{5}$$

where $\boldsymbol{1}$ is a $K$-dimensional column vector with 1 in each element. Proof can be found in Appendix A. The key idea of the proof is to not rely on the condition $\sum_{k=1}^K \overline{\ell}(k, g) = 1$ used in Ishida et al. (2017), which is a condition inspired by the property of binary 0-1 loss $\ell_{0-1}$, where $\ell_{0-1}(z)$ is 1 if $z < 0$ and 0 otherwise. Note that such a technique was also used when designing unbiased risk estimators for learning from positive and unlabeled data in a binary classification setup (**?**), but was later shown to be unnecessary (du Plessis et al., 2015).

According to Theorem 1, we can derive an equivalent form,

$$\overline{\ell}(k, g) = -(K-1) \cdot \ell(k, g) + \sum_{j=1}^K \ell(j, g). \tag{6}$$

Therefore, the classification risk can be written as

$$R(g; \ell) = \sum_{k=1}^K \overline{\pi}_k \mathbb{E}_{\overline{P}_k} \Big[ -(K-1) \cdot \ell(k, g) + \sum_{j=1}^K \ell(j, g) \Big]. \tag{7}$$

This expression of the classification risk allows us to naively approximate it in an unbiased fashion using complementarily labeled data as

$$\widehat{R}(g; \ell) = \sum_{k=1}^K \frac{\widehat{\overline{\pi}}_k}{n_k} \sum_{i=1}^{n_k} \Big[ -(K-1) \cdot \ell\big(k, g(\boldsymbol{x}_i)\big) + \sum_{j=1}^K \ell\big(j, g(\boldsymbol{x}_i)\big) \Big], \tag{8}$$

where $n_k$ is the number of samples complementarily labeled as the $k$th class. It is worth noting that, in the above derivation, there are no constraints on the loss function and classifier. Thus, we can use any convex/non-convex loss and any linear/non-linear parametric/non-parametric model for complementary learning.

### 3.2 NECESSITY OF RISK CORRECTION

The original expression of the classification risk (1) includes an expectation over non-negative loss $\ell : [K] \times \mathbb{R}^K \to \mathbb{R}_+$, so the risk and its empirical approximator are both lower-bounded by zero. On the other hand, the expression (7) derived above contains an negative element. Although (7) is still non-negative by definition, due to the negative term, its empirical estimator can go negative, leading to over-fitting.

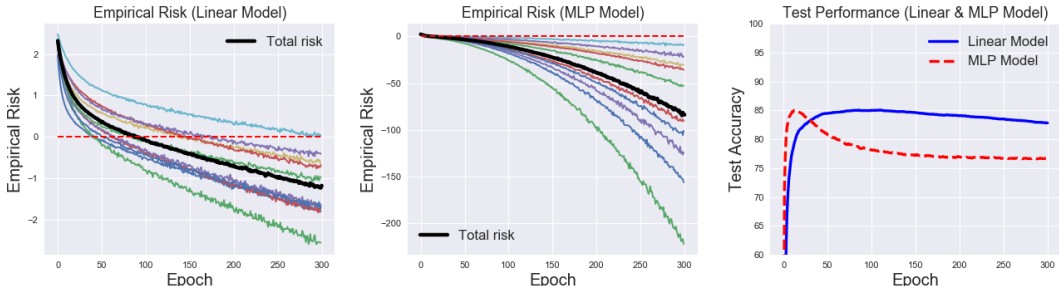

**Figure 1:** The left and middle graphs shows the total risk (8) (in black color) and the risk decomposed into each *ordinary* class term (9) (in other colors) for training data with linear and MLP models, respectively. As an MLP model, a one-hidden-layer neural network with $500$ units was used, with *ReLU* (Nair & Hinton, 2010) as the activation function, *Adam* (Kingma & Ba, 2015) for optimization with learning rate $5e-5$ and weight decay of $1e-4$. The right graph shows the corresponding test accuracy for both models.

We elaborate on this issue with an illustrative numerical example. In the left graph of Figure 1, we show an example of training a linear model trained on the handwritten digits dataset MNIST[1], with complementary labels generated to satisfy (3). We used *Adam* (Kingma & Ba, 2015) for optimization with learning rate $5e-5$, and weight decay of $1e-4$ with 300 epochs. The empirical classification risk (8) is shown in black. We can see that the empirical classification risk continues decreasing and can go below zero at around 100 epochs. The test accuracy on the right graph hits the peak also at around epoch 100 and then the accuracy gradually deteriorates.

This issue stands out even more significantly when we use a flexible model. The middle graph shows the empirical classification risk for a multilayer perceptron (MLP) with one hidden layer (500 units), where *ReLU* (Nair & Hinton, 2010) was used as the activation function. The optimization setup was the same as the case of the linear model above. We can see the empirical risk decreasing much more quickly and going negative. Correspondingly, as the right graph shows, the test accuracy drops significantly after the empirical risk goes negative.

In fact, a similar issue has already been conceivable in the original paper by Ishida et al. (2017): According to Theorem 1 in Ishida et al. (2017), the unbiased risk estimator includes subtraction of a positive constant term which increases with respect to the number of classes. This means that the learning objective of Ishida et al. (2017) has a (negative) lower bound. Our objective, however, is unbounded from below and thus can end up in even heavier overfitting.

### 3.3 NON-NEGATIVE RISK ESTIMATOR

As we saw in Section 3.2, our risk estimator can suffer from overfitting due to the non-negative issue. Here, we propose a correction to the risk estimator to overcome this problem.

Each term in the risk with ordinary labels (right-hand side of (2)), which corresponds to each class, is non-negative. We can reformulate (7) in order to show the counterpart for each non-negative term in right-hand side of (2) for complementarily labeled data as

$$R(g;\ell) = \sum_{k=1}^{K} \overline{\pi}_k \Big[ -(K-1) \cdot \mathbb{E}_{\overline{P}_k}[\ell(k,g)] + \sum_{j=1}^{K} \mathbb{E}_{\overline{P}_j}[\ell(k,g)] \Big]. \tag{9}$$

These counterparts (9) were originally non-negative when ordinary labels were used. In the left and middle graphs of Figure 1, we plot the decomposed risk with respect to each *ordinary* class (9) (shown in different colors). We can see that the decomposed risks for all classes become negative eventually. Based on this observation, our basic idea for correction is to enforce non-negativity for each ordinary class, with the expression based on complementary labels. More specifically, we propose a non-negative (nn) version by

$$R_{\text{nn}}(g;\ell) = \sum_{k=1}^{K} \max \Big\{ 0, \overline{\pi}_k \Big[ -(K-1) \cdot \mathbb{E}_{\overline{P}_k}[\ell(k,g)] + \sum_{j=1}^{K} \mathbb{E}_{\overline{P}_j}[\ell(k,g)] \Big] \Big\}. \tag{10}$$

---

[1]See http://yann.lecun.com/exdb/mnist/.

This non-negative risk can be naively approximated by the sample average as

$$\widehat{R}_{\text{nn}}(g;\ell) = \sum_{k=1}^{K} \max\Big\{0, \overline{\pi}\Big[ -\frac{K-1}{n_k}\sum_{i=1}^{n_k} \ell(k, g(\boldsymbol{x}_i)) + \sum_{j=1}^{K}\frac{1}{n_{i'}}\sum_{i'=1}^{n_{i'}} \ell(j, g(\boldsymbol{x}_{i'}))\Big]\Big\}. \quad (11)$$

Enforcing the reformulated risk to become non-negative was previously explored in Kiryo et al. (2017), in the context of binary classification from positive and unlabeled data. The positive class risk is already bounded below by zero in their case (because they have true positive labels), so there was a max operator only on the negative class risk. We basically follow their footsteps, but since our setting is a multi-class scenario and also differs by not having *any* true labels, we put a max operator on every $K$ class.

### 3.4 IMPLEMENTATION

**Implementation with max operator**   We show practical implementation under stochastic optimization for our non-negative risk estimator. An unfortunate issue is that the minimization of (11) is not point-wise due to the max-operator, thus cannot be used directly for stochastic optimization methods with mini-batch. However, an upper bound of the risk can be minimized in parallel by using mini-batch as the following,

$$\frac{1}{N}\sum_{i=1}^{N}\sum_{k=1}^{K} \max\Big\{0, \overline{\pi}_k\Big[ -(K-1)\widehat{\mathbb{E}}_{\overline{P}_k}[\ell(k,g); \mathcal{X}_{\overline{k}}^i] + \sum_{j=1}^{K}\widehat{\mathbb{E}}_{\overline{P}_j}[\ell(k,g); \mathcal{X}_{\overline{j}}^i]\Big]\Big\}, \quad (12)$$

where $\widehat{\mathbb{E}}$ is the empirical version of the expectation and $\mathcal{X}_{\overline{j}}^i$ denotes the samples complementarily labeled as the $j$th class in the $i$th mini-batch.

**Implementation with gradient ascent**   If the objective is negative for a certain mini-batch, the previous implementation based on the max operator will avoid the objective to further *decrease*. However, if the objective is already negative, that mini-batch has already started to overfit. Therefore, it would be preferable to *increase* itself to make this mini-batch less overfitted.

Our idea is the following. We denote the risk that corresponds to the $k$th ordinary class for the $i$th mini-batch as

$$r_k^i(\theta) = \overline{\pi}_k\Big[ -(K-1)\widehat{\mathbb{E}}_{\overline{P}_k}[\ell(k,g); \mathcal{X}_{\overline{k}}^i] + \sum_{j=1}^{K}\widehat{\mathbb{E}}_{\overline{P}_j}[\ell(k,g); \mathcal{X}_{\overline{j}}^i]\Big], \quad (13)$$

and the total risk as $L^i(\theta) = \sum_{k=1}^{K} r_k^i(\theta)$. When $\min_k\{r_k^i(\theta)\}_{k=1}^K \geq -\beta$, we conduct gradient descent as usual with gradient $\nabla_\theta L^i(\theta)$. On the other hand, if $\min_k\{r_k^i(\theta)\}_{k=1}^K < -\beta$, we first squash the class-decomposed risks over $-\beta$ to $-\beta$ with a min operator, and then sum the results: $\widetilde{L}^i(\theta) = \sum_{k=1}^K \min\{-\beta, r_k^i(\theta)\}$.

Next we set the gradient in the opposite direction with $-\nabla_\theta \widetilde{L}^i(\theta)$. Conceptually, we are going *up* the gradient $\nabla_\theta \widetilde{L}^i(\theta)$ for *only* the class-decomposed risks below $-\beta$, to avoid the class-decomposed risks that are already large to further increase. Note that $\beta$ is a hyper-parameter that controls the tolerance of negativity. $\beta = 0$ would mean there is zero tolerance, but in practice we can also have $-\beta \neq 0$ for a threshold that allows some negative ($-\beta < 0$) or positive ($-\beta > 0$) amount. The procedure is shown in detail in Algorithm 1.

## 4 EXPERIMENTS

In this section, we experimentally compare our three proposed methods (Algorithm 1, (8) and (12), with two baseline methods from Ishida et al. (2017) and Yu et al. (2018). Table 2 describes the summary statistics of the benchmark datasets used in this section. The implementation is based on Pytorch[2] and our code will be available on http://anonymized for reproducing results.

---
[2] https://pytorch.org

---

**Algorithm 1** Proposed algorithm with gradient ascent

---

**Input:** complementarily labeled training data $\{\mathcal{X}_{\overline{k}}\}_{k=1}^{K}$, where $\mathcal{X}_{\overline{k}}$ denotes the samples complementarily labeled as class $\overline{k}$;
**Output:** model parameter $\theta$ for $g(\boldsymbol{x}; \theta)$
1: Let $\mathcal{A}$ be an external SGD-like stochastic optimization algorithm such as Kingma & Ba (2015)
2: Denote $\{\mathcal{X}_{\overline{j}}^{i}\}$ as the $i$-th mini-batch for complementary class $j$
3: Denote $L^{i}(\theta) = \sum_{k=1}^{K} r_{k}^{i}(\theta)$
4: Denote $r_{k}^{i}(\theta) = \overline{\pi}_{k}\big[ -(K-1)\widehat{\mathbb{E}}_{\overline{P}_{k}}[\ell(k,g); \mathcal{X}_{\overline{k}}^{i}] + \sum_{j=1}^{K} \widehat{\mathbb{E}}_{\overline{P}_{j}}[\ell(k,g); \mathcal{X}_{\overline{j}}^{i}]\big]$
5: Denote $\widetilde{L}^{i}(\theta) = \sum_{k=1}^{K} \min\{-\beta, r_{k}^{i}(\theta)\}$
6: **while** no stopping criterion has been met:
7:     Shuffle $\{\mathcal{X}_{\overline{j}}\}_{\overline{j}}^{K}$ into $N$ mini-batches;
8:     **for** $i = 1$ **to** $N$:
9:         **if** $\min_{k}[r_{1}^{i}(\theta), \ldots, r_{k}^{i}(\theta), \ldots, r_{K}^{i}(\theta)] > -\beta$:
10:             Set gradient $\nabla_{\theta} L^{i}(\theta)$;
11:             Update $\theta$ by $\mathcal{A}$ with its current step size $\eta$;
12:         **else**:
13:             Set gradient $-\nabla_{\theta}\widetilde{L}^{i}(\theta)$;
14:             Update $\theta$ by $\mathcal{A}$ with a discounted step size $\gamma\eta$;

---

### 4.1 SETUP

For MNIST and Fashion-MNIST, a linear-in-input model with a bias term and a MLP model ($d - 500 - 1$) was trained with softmax cross-entropy loss function. Weight decay of $1e-4$ for weight parameters and learning rate of $5e-5$ for Adam (Kingma & Ba, 2015) was used.

For CIFAR-10, DenseNet (Huang et al., 2017) and Resnet-18 (He et al., 2016) with default parameter settings were trained. Weight decay of $5e-4$ and initial learning rate of $1e-2$ was used. For optimization, stochastic gradient descent was used with the momentum set to 0.9. Learning rate was halved every 30 epochs.

We trained and compared 5 methods (*Free* (8), *Max operator* (12), *Gradient ascent* (Alg.1), *PC* (Ishida et al., 2017) and *Forward* (Yu et al., 2018)) with only complementarily labeled data. Note that the first three are the proposed methods. We complementarily labeled our benchmark datasets so that the assumption of (3) is satisfied. This is straightforward when the dataset has a uniform (ordinarily-labeled) class prior, because it reduces to just choosing a class randomly other than the true class. For *Gradient ascent*, we used $\beta = 0$ and $\gamma = 0$ for simplicity. We trained 300 epochs, where mini-batch was set to 100.

### 4.2 RESULTS

Instead of showing the test accuracy for a single chosen model based on validation, we show the accuracy for all 300 epochs on test data to demonstrate how the issues discussed in Section 3.2 appear and how different implementations Section 3.4 is effective. In Figure 2, we show the mean test accuracy and standard deviation for 4 trials for the three benchmark datasets, on test data evaluated with ordinary labels.

First we compare our three proposed methods with each other. For linear models in MNIST and Fashion-MNIST, all proposed methods work similarly. However in the case of using a more flexible model (MLP model for MNIST/Fashion-MNIST, Densenet/Resnet for CIFAR-10), we can see that *Free* is the worst, *Max operator* is better and *Gradient ascent* is the best out of the proposed three methods at the end of all epochs (*Free* < *Max operator* < *Gradient ascent*). These results are consistent with the discussions of overfitting in Section 3.2 and the motivations for different implementations in Section 3.4.

Next, we compare with baseline methods. For linear models, all methods have similar performance. However for deep models, the superiority stands out for *Gradient ascent* for all datasets.

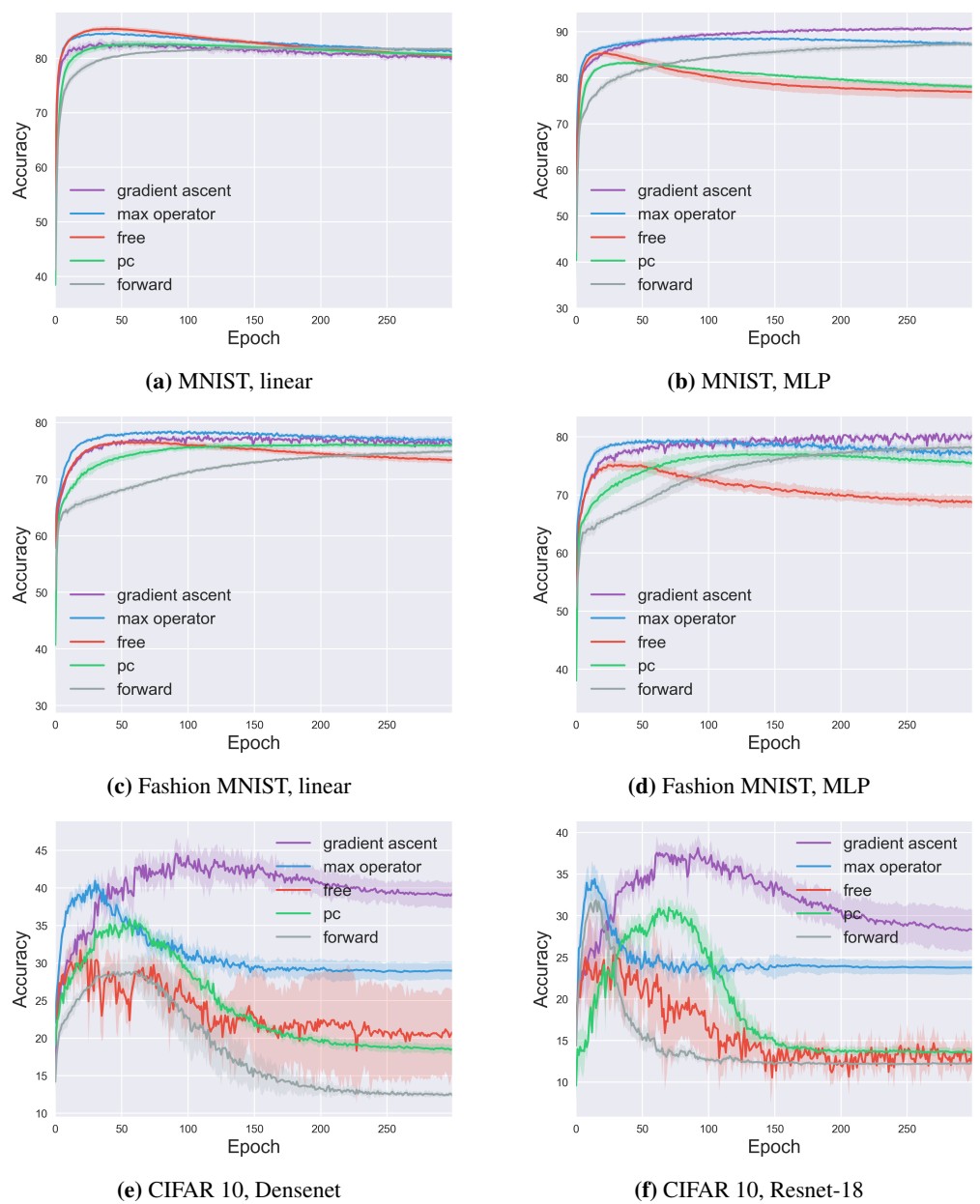

**Figure 2:** Experimental results for various datasets and models. Dark colors show the mean accuracy of 4 trials and light colors show standard deviation.

## 5 CONCLUSION

We first proposed a general risk estimator for learning from complementary labels that does not require restrictions on the form of the loss function or the model. However, since the proposed method suffers from overfitting, we proposed a modified version to alleviate this issue in two ways and have better performance. At last, we conducted experiments to show our proposed method outperforms or is comparable to current state-of-the-art methods for various benchmark datasets and for both linear and deep models.

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

## A    PROOF OF THEOREM 1

*Proof.* First of all,

$$\mathbb{P}(X, \overline{Y} = \overline{y}) = \frac{1}{K-1} \sum_{y \neq \overline{y}} \mathbb{P}(X, Y = y) \tag{14}$$

$$= \frac{1}{K-1} \Big( \sum_{y=1}^{K} \mathbb{P}(X, Y = y) - \mathbb{P}(X, Y = \overline{y}) \Big) \tag{15}$$

$$= \frac{1}{K-1} \big( \mathbb{P}(X) - \mathbb{P}(X, Y = \overline{y}) \big). \tag{16}$$

The first equality holds since the marginal distribution is equivalent for $\mathcal{D}$ and $\overline{\mathcal{D}}$ and we assume (3). Consequently,

$$\mathbb{P}(\overline{Y} = \overline{y}|X = x) = \frac{\mathbb{P}(X = x, \overline{Y} = \overline{y})}{\mathbb{P}(X = x)} \tag{17}$$

$$= \frac{1}{K-1} \cdot \Big( 1 - \frac{\mathbb{P}(X, Y = \overline{y})}{\mathbb{P}(X = x)} \Big) \tag{18}$$

$$= \frac{1}{K-1} \cdot \big( 1 - \mathbb{P}(Y = \overline{y}|X = x) \big) \tag{19}$$

$$= -\frac{1}{K-1} \mathbb{P}(Y = \overline{y}|X = x) + \frac{1}{K-1}. \tag{20}$$

More simply, we have

$$\boldsymbol{\eta}(x) = -(K-1)\overline{\boldsymbol{\eta}}(x) + \mathbf{1}. \tag{21}$$

Finally, we transform the classification risk,

$$R(g; \ell) = \mathbb{E}_{(X,Y) \sim \mathcal{D}}[\ell(Y, \boldsymbol{g}(X))] \tag{22}$$

$$= \mathbb{E}_{X \sim M}[\boldsymbol{\eta}^\top \boldsymbol{\ell}(\boldsymbol{g}(X))] \tag{23}$$

$$= \mathbb{E}_{X \sim M}\big[ \big( -(K-1)\overline{\boldsymbol{\eta}}^\top + \mathbf{1}^\top \big) \boldsymbol{\ell}(\boldsymbol{g}(X)) \big] \tag{24}$$

$$= \mathbb{E}_{X \sim M}\big[ -(K-1)\overline{\boldsymbol{\eta}}^\top \boldsymbol{\ell}(\boldsymbol{g}(X)) + \mathbf{1}^\top \boldsymbol{\ell}(\boldsymbol{g}(X)) \big] \tag{25}$$

$$= \mathbb{E}_{(X,\overline{Y}) \sim \overline{\mathcal{D}}}\big[ -(K-1) \cdot \ell(\overline{Y}, \boldsymbol{g}(X)) \big] + \mathbf{1}^\top \mathbb{E}_{X \sim M}\big[ \boldsymbol{\ell}(\boldsymbol{g}(X)) \big] \tag{26}$$

$$= \sum_{k=1}^{K} \mathbb{E}_{X \sim \overline{P}_k}\Big[ \overline{\pi}_k \cdot \Big( -(K-1) \cdot \ell(k, \boldsymbol{g}(X)) + \mathbf{1}^\top \boldsymbol{\ell}(\boldsymbol{g}(X)) \Big) \Big] \tag{27}$$

$$= R(g; \overline{\ell}) \tag{28}$$

for the complementary loss,

$$\overline{\ell}(k, \boldsymbol{g}) := -(K-1)\ell(k, \boldsymbol{g}) + \mathbf{1}^\top \boldsymbol{\ell}(\boldsymbol{g}), \tag{29}$$

which concludes the proof. □

## B    DETAILS OF DATASETS USED IN SECTION 4

In Table 2, we explain the details of the datasets used in Section 4. See http://yann.lecun.com/exdb/mnist/ for MNIST, https://github.com/zalandoresearch/fashion-mnist for Fashion-MNIST, and https://www.cs.toronto.edu/~kriz/cifar.html for CIFAR-10.

**Table 2:** Summary statistics of benchmark datasets.

| Name | # Train | # Test | # Dim | # Classes | Model |
|---|---|---|---|---|---|
| MNIST | 60,000 | 10,000 | 784 | 10 | Linear, MLP |
| Fashion MNIST | 60,000 | 10,000 | 784 | 10 | Linear, MLP |
| CIFAR-10 | 60,000 | 10,000 | 2,048 | 10 | DenseNet, Resnet |

