# OpenReview forum: "Complementary-label learning for arbitrary losses and models"
_ICLR.cc/2019/Conference_

### Official Review · AnonReviewer3 · 2018-11-02
**Well written paper about an interesting problem. The major problem is that the core part of the contribution is a special case of previously published more general framework not cited in the paper.**

**Rating:** 6
**Confidence:** 4

**Review:**

pros:

- Clearly written and sound paper.
- Addresses interesting problem.
- Improves existing methods used for this learning scenario.

cons:

- The core contribution is a special case of previously published more general framework which is not cited in the paper.

It is clearly written paper with a good motivation. The major problem is that the core contribution, namely, the risk reformulation in Theorem 1 and the derived loss (6), are special cases of more general framework published in
   Jesus Cid-Sueiro et al. Consistency of Losses for Learning from Weak Labels. ECML 2014.

The work of [Cid-Sueiro2014] proposes a general way how to construct losses for learning from weak labels. They require that the distribution of weak labels is a linear transformation of the true label distribution, i.e. the assumption (3) of the paper under review. According to [Cid-Sueiro2014], the loss on weak labels is constructed by $weak_loss = L*original_loss$, where $L$ is the left inversion of the "mixing matrix" $T$ in (3). [Cid-Sueiro2014] also shows that such weak loss is classification calibrated which implies statistical consistency of the method.

Learning from complementary labels is a special case when the mixing matrix is $T=(E-I)/(K-1)$ (E is unitary matrix, I is matrix of ones, K is number of labels). In this case, the left inversion of $T$ is simply $L=- E*(K-1) + I$ and so the weak loss is $weak_loss=L*loss$ which corresponds to the loss (5) proposed in the paper under review (in fact, the loss (5) also adds a constant term (Y-2)/(Y-1) which however has no effect on the minimizer).

The novel part of the paper is the non-negative risk estimator proposed in sec 3.3 and the online optimization methods addressed in sec 3.4. These extensions, although relatively straightforward, are empirically shown to significantly improve the results.

---

> ### Author Response · Authors · 2018-11-26
> **Thank you for your reviews!**
>
> Thank you very much for the insightful reviews!
>
> Q) Theorem 1 and derived loss are special cases of more general framework published in prior work.
> A) Thank you very much for pointing this out and explaining the relationship between our paper.  We would like to clarify our contributions carefully and explain the relationship in our paper.

---

### Official Review · AnonReviewer2 · 2018-11-03
**Clear paper on interesting setup, but claims are undermined by issues with first estimator + lack of motivation for assumptions**

**Rating:** 5
**Confidence:** 4

**Review:**

This paper proposes an improved approach to the "complementary-label" form of weak supervision, in which a label that is *not* the true label is marked. Specifically, this paper proposes an unbiased estimator that accepts arbitrary loss functions and models. Noting that this proposed estimator can suffer from overfitting due to unbounded negative loss, a lower-bounded estimator is proposed. Experiments are then performed on several image classification datasets.

Pros:
- This paper addresses a creative form of weak supervision, proposed by prior work, in which labels that are *not* the true label are labeled, in a clear fashion.

- The first proposed estimator is unbiased, as shown by a proof, and accepts arbitrary losses, an improvement over prior approaches

- The overall presentation is clear and clean

Cons:
- One of the main claims of the paper is the proposal of an unbiased estimator. However, this estimator then does not seem to work well enough due to degenerate negative loss.  So then a modified version is proposed- which does not appear to be unbiased?  Either way, no assertion or proof of it being unbiased is given.  So then presumably this also reverses the claim of being able to cross-validate?  This seems like a major weakening of the paper's contributions

- Since the unbiased estimator does not appear to work well, two implementations of a corrected one are proposed, using heuristic approaches without explicit theoretical guarantees.  This shifts the burden to the experimental studies.  These are somewhat thorough, but not extremely so: for example, one set of hyperparameters were used for all of the methods?  This seems like it could implicitly handicap / favor some over others?

- The proposed estimator is based on the assumption that the probability of classes in the complement set (the set of labels other than the one marked as incorrect) is uniformly distributed (e.g. see beginning of Proof of thm 1).  However, this seems like a potentially naive assumption. Indeed, in the related work section, it is mentioned that work in 2018 already considered the case where this uniformity assumption does not hold.

- More broadly, but following from the above: The paper does not provide any real world examples, real or hypothetical, to give the reader an idea of whether the above uniformity assumption---or really any of these assumptions---are well-motivated or empirically justified.  At the bottom of page 3 in the related work, a concrete application used in prior work is mentioned---where crowd workers are shown single labels and vote Y/N, leading to a mix of standard (if Y) and complement-labeled (if N) data---however this mixed setting is not considered explicitly in this paper.  So, how is the reader supposed to get any idea of whether the assumed setup is motivated or justified?  The experiments do not provide this, because the complementary labels are synthetically generated according to the model assumed in the paper.  Additionally, it is briefly mentioned that collecting complementary labeled data is faster, but again no concrete examples are given to support this.

---

> ### Author Response · Authors · 2018-11-26
> **Thank you for your reviews!**
>
> Thank you very much for the review and for the important questions!
>
> Q) Modified version appears to be biased, and does this reverse the claim of being able to cross-validate?
> A) We apologize for the lack of clarity in the paper, but even if our learning objective uses a modified version that can potentially be a biased estimator, we can still use an unbiased version for our cross-validating objective.  We would like to demostrate a very simple example (with a single validation split).  We report the classification accuracy with Fashion-MNIST with just one trial: Free(proposed):59.06% / gradient-ascent(proposed):79.58% / forward[Xiyu'18]:46.72% / pairwise-comparison[Ishida'17]:76.03%.  9 hyper-parameter candidate combinations of weight decay (1e-4, 1e-5, 1e-6) and learning rate (1e-4, 1e-5, 1e-6) were used with 150 epochs.  SGD with momentum 0.9 was used for optimization.  We reported the test accuracy of the best model based on validation score (calculated from only complementarily-labeled data) from all epochs and all hyper-parameter combinations.  For forward method [Xiyu'18], we simply used their proposed learning objective for validation criteria.  The model was multi-layer perceptron with d-50-1 (In Figure 2, d-500-1 was used).  Due to the time constraint, we were able to only finish a very simple setup with a single trial, with very few hyper-parameters and few candidates for each of them, so our main message here is not the result itself (for example “forward” is too weak and we can guess optimal hyper-parameters were not included), but to show that validation is possible with our unbiased estimator.  We would like to report results for extensive experiments in the final version.
>
> Q) The hyper-parameters are fixed.  Will this implicitly handicap / favor some over others?
> A) Our motivation of the experiments was to demonstrate the failure of the proposed method based on Theorem 1, and how the two modifications solve the overfitting issues and show test results of all epochs during training.  However, as you point out, this is not a good demonstration of comparing with the best hyper-parameter for each method, so we showed some simple experimental results that tune hyper-parameters with (complementary labeled) validation data in the answer to your previous question.  We will add more experiments to demonstrate this.
>
> Q) Uniform assumption is naïve.
> A) A potentially biased (but consistent) method has already been proposed for a non-uniform assumption [22], but one of our future work is to explore if proposing a non-uniform version of the unbiased estimator is possible or not.
>
> Q) The only justification of the uniform assumption is the mixed setting of crowdsourcing, but there is no mixed setting in experiments.
> A) We would like to demonstrate with more experiments using the mixed setting with both ordinary and complementary labels.

---

### Official Review · AnonReviewer1 · 2018-11-09
**Interesting setting, but problems with the original estimator and limited experimental evaluation weaken the claims**

**Rating:** 5
**Confidence:** 3

**Review:**

Pros:
- The authors consider an interesting problem of learning from complementary labels
- They propose an approach that, assuming that the complementary label is selected uniformly at random, provides an unbiased estimate for any loss function, which is an improvement over the previous work.
- Experiments show promising results for modifications of the proposed estimate

Cons:
- Having an unbiased estimate doesn't imply that its minimisation is a successful learning strategy. Indeed, the authors show that minimising their original estimate for the cross-entropy loss leads to overfitting. While the authors attribute this behaviour to the fact that the estimate can be negative, I believe the loss being negative is not problem per se (for example, substituting 0/1 loss with -100/-99 loss would not change the learning; similarly, this is not a problem for the losses considered in [Ishida'17]). I would rather attribute the problem to the fact that the proposed estimate is unbounded from below and there are no generalisation guarantees for it. Indeed, assuming there exists a training example that appears in the training set only once, with one complementary label, estimate (8) can be made arbitrary small by just training to predict probability 0 for the provided complementary label on that example ( and any non-zero probability for other classes).
- to cope with the above mentioned problem, the authors propose two heuristic-based modifications of the estimate, which are potentially biased. This weakens the initial motivation for finding an unbiased estimate and shifts the focus towards the experimental evaluation
- one of the mentioned motivations for unbiased estimates - being able to perform model selection on complementary labeled validation set - is not illustrated in the experiments

Questions:
- I believe 1/(K-1) normalisation factor in (5) is not needed
- there seems to be a mistake in (9) (and its modifications later on) - I would expect either the subscript $j$ of the probability distribution in the last summand to be exchanged with $k$ in the loss, or a factor $\pi_j/\pi_k$ added
- also, I think there are some mistakes in subscripts in (11)
- what loss is the method from [Ishida'17] optimising in the experiments?

---

> ### Author Response · Authors · 2018-11-26
> **Thank you for your reviews!**
>
> Thank you very much for reading our paper in depth and for your reviews!
>
> Q) About the reasons for overfitting.
> A) Thank you for the valuable feedback.  A possible empirical test would be to make a non-negative version of [Ishida’17] and compare it with the original estimator in [Ishida’17].
>
> Q) The modifications of the unbiased estimator lead to biased estimators.
> A) Yes, this is true, but we would like to point out that even if our learning objective is biased (with either non-negative version or gradient ascent version), we can still use the unbiased version for our cross-validating objective.  Therefore, performing cross validation with only complementary data can still be achieved.  Since we did not demonstrate the validation procedure in our experiments, we showed some preliminary results for experiments with validation in the reply to Reviewer2.
>
> Q) About the three mistakes on the notations of equations.
> A) Thank you very much for pointing this out.  We will fix these issues in our final version.
>
> Q) What is the loss from [Ishida’17]?
> A) We used pairwise comparison multi-class loss with sigmoid binary loss, which was used in the experimental section of [Ishida’17].

---

### Meta-Review · Area_Chair1 · 2018-12-13

**Confidence:** 5
**Recommendation:** Reject

**Metareview:**

The paper studies learning from complementary labels – the setting when example comes with the label information about one of the classes that the example does not belong to. The paper core contribution is an unbiased risk estimator for arbitrary losses and models under this learning scenario, which is an improvement over the previous work, as rightly acknowledged by R1 and R2.

The reviewers and AC note the following potential weaknesses: (1) R3 raised an important concern that the core technical contribution is a special case of previously published more general framework which is not cited in the paper. The authors agree with R3 on this matter; (2) the proposed unbiased estimator is not practical, e.g. it leads to overfitting when the cross-entropy loss is used, it is unbounded from below as pointed out by R1; (3) the two proposed modifications of the unbiased estimator are biased estimators, which defeats the motivation of the work and limits its main technical contributions; (4) R2 rightly pointed out that the assumption that complementary label is selected uniformly at random is unrealistic – see R2’s suggestions on how to address this issue.
While all the reviewers acknowledged that the proposed biased estimators show advantageous performance on practice, the AC decides that in its current state the paper does not present significant contributions to the prior work, given (1)-(3), and needs major revision before submitting for another round of reviews.